# Guided accumulation of active particles by topological design of a second-order skin effect

Lucas S. Palacios[1], Serguei Tchoumakov[2], Maria Guix [1], Ignacio Pagonabarraga [3,4,5], Samuel Sánchez [1,6✉] & Adolfo G. Grushin [2✉]

Collective guidance of out-of-equilibrium systems without using external fields is a challenge of paramount importance in active matter, ranging from bacterial colonies to swarms of self-propelled particles. Designing strategies to guide active matter and exploiting enhanced diffusion associated to its motion will provide insights for application from sensing, drug delivery to water remediation. However, achieving directed motion without breaking detailed balance, for example by asymmetric topographical patterning, is challenging. Here we engineer a two-dimensional periodic topographical design with detailed balance in its unit cell where we observe spontaneous particle edge guidance and corner accumulation of self-propelled particles. This emergent behaviour is guaranteed by a second-order non-Hermitian skin effect, a topologically robust non-equilibrium phenomenon, that we use to dynamically break detailed balance. Our stochastic circuit model predicts, without fitting parameters, how guidance and accumulation can be controlled and enhanced by design: a device guides particles more efficiently if the topological invariant characterizing it is non-zero. Our work establishes a fruitful bridge between active and topological matter, and our design principles offer a blueprint to design devices that display spontaneous, robust and predictable guided motion and accumulation, guaranteed by out-of-equilibrium topology.

[1] Institute for Bioengineering of Catalonia (IBEC), Barcelona Institute for Science and Technology (BIST), Barcelona, Spain. [2] University of Grenoble Alpes, CNRS, Grenoble INP, Institut Néel, Grenoble, France. [3] Departament de Física de la Matèria Condensada, Universitat de Barcelona, Barcelona, Spain. [4] University of Barcelona Institute of Complex Systems (UBICS), Universitat de Barcelona, Barcelona, Spain. [5] CECAM, Centre Européen de Calcul Atomique et Moléculaire, École Polytechnique Fédérale de Lausanne (EPFL), Lausanne, Switzerland. [6] Institució Catalana de Recerca i Estudis Avançats (ICREA), Barcelona, Spain. ✉email: ssanchez@ibecbarcelona.eu; adolfo.grushin@neel.cnrs.fr

The last decades of research in condensed matter physics have revealed that exceptionally robust electronic motion occurs at the boundaries of a class of insulators known as topological insulators[1,2]. These ideas extend beyond solid-state physics, and predict guided boundary motion in systems including photonic[3], acoustic, and mechanical systems[4].

The recent discovery that topological properties emerge in the class of out-of-equilibrium systems described by non-Hermitian matrices, which includes active matter systems[5], has opened the possibility to engineer robust behavior out of equilibrium[6,7]. While in equilibrium topological boundary states are predicted by a non-zero bulk topological invariant, a feature known as the bulk-boundary correspondence, in non-Hermitian systems, this correspondence is broken by the skin effect[8–13]. For example, in a one-dimensional (1D) chain of hopping particles, the first-order non-Hermitian skin effect arises from the asymmetry between left and right hopping probabilities, which results in an accumulation of a macroscopic number of modes, of the order of the system size, on one side of the system. This 1D effect occurs in systems without an inversion center, and has been observed in photon dynamics[14], mechanical metamaterials[15–17], optical fibers[18], and topoelectrical circuits[19,20]. In 1D, the skin effect occurs if a topological invariant, the integer associated with the winding of the complex spectrum of the normal modes, is non-zero[21–23].

Higher-dimensional versions of the skin effect can display a considerably richer and subtle phenomenology[24–36]. In this work we are interested in an elusive second-order non-Hermitian skin effect, predicted only in out-of-equilibrium systems in two dimensions (2D)[30–35]. It differs from the first-order skin effect because (i) it can occur in inversion symmetric systems, accumulating modes at opposing corners rather than edges[20,37], and (ii) the number of accumulated modes is of the order of the system boundary $L$, rather than its area $L^2$. While the first-order non-Hermitian skin effect requires inversion to be broken, e.g., due to an applied field, the emergence of the second-order non-Hermitian skin effect is guaranteed by the presence of certain symmetries[33,34]. However, predicting the second-order non-Hermitian skin effect is challenging in general, and it remains unobserved. Dissipation, which drives a system out of equilibrium, is hard to control experimentally in quantum electronic devices, therefore calling for platforms to realize the second-order non-Hermitian skin effect.

Active matter systems[38] are a natural platform to explore non-Hermitian topological physics, since these systems absorb and dissipate energy[5]. Often, the hydrodynamic equations that describes their flow can be mapped to a topological Hamiltonian. This strategy predicts topologically protected motion of topological waves in active-liquid metamaterials[39–41], skin-modes in active elastic media[37], and emergent chiral behavior for periodic arrays of defects[42]. Non-Hermitian topology in active matter has been demonstrated experimentally in active nematic cells[43], and robotic[16] and piezoelectric metamaterials[17].

In this work, we design microfabricated devices[44–46] that display a controllable second-order non-Hermitian skin effect (see Fig. 1). These devices are designed to satisfy detailed balance on their unit cell, such that the flow of particles through a unit cell vanishes. The non-Hermitian skin effect dynamically breaks this detailed balance on the top and bottom edges, and we use it to guide and accumulate self-propelled Janus particles. In contrast to hydrodynamic descriptions, the topological particle dynamics in our devices is quantitatively described by a stochastic circuit model[47] without fitting parameters. It establishes that topological circuits, where a topological invariant $\nu = 1$, display the second-order non-Hermitian skin effect that guides and accumulates particles more efficiently than the topologically trivial circuits, with $\nu = 0$. This phenomenon occurs without external stimuli, e.g., electrical or magnetic fields, a useful feature for active matter applications[38,48,49], and to extend our design principles to metamaterial platforms[3,4,50].

## Results

**Coupled-wire device design and stochastic model**. Our design realizes the coupled-wire construction, a theoretical tool to construct topological phases[51]. This is possible by the precise engineering of microchannel devices (see the "Methods" section)[44]. Each device contains two types of horizontal microchannels, the wires, which are coupled vertically, forming a 2D mask (see Fig. 1a, b). The horizontal microchannels are consecutive left or right oriented hearth-shaped ratchets, that favor a unidirectional motion toward their tip[46]. Their left–right orientation alternates vertically. These horizontal microchannels are coupled by vertical microchannels that are straight, designed to imprint a symmetric vertical motion to the nanoparticles. The vertical microchannels alternate in width, with successive narrow and wide channels. Wide channels are more likely to be followed by the active particles than narrow channels.

Using these principles, we design two types of devices that we coin trivial and topological, depicted in Fig. 1a, b, respectively. They only differ in that the narrow and wide channels exchange their roles along the vertical direction (see central inset of Fig. 1a, b). Both trivial and topological designs have the same number of left and right oriented ratchets, so they satisfy global balance.

Active particles are injected within the device and move along the microchannels walls, see Fig. 1c, d and "Methods". We track the particles from recorded videos with an in-house developed software based on a tiny YOLOv3 neuronal network[52]. We locate the particles within a grid of regularly spaced cells (dashed lines in Fig. 1e). We use the recorded trajectories as a database, which we can post-select and synchronize in order to study the average particle motion starting from various initial configurations (see "Methods"). In Fig. 1f, g, h, we show different initial distributions of particles for latter use. Figure 1f shows the particle distributions when all recorded trajectories are synchronized to start at the same initial time. We observe more particles at the borders than in the bulk because of the particle flux from outside the device. Since this flux cannot be controlled, we cannot directly choose the initial particle distribution. We overcome this limitation by post-selecting trajectories so that they start either from the top left and bottom right cells (Fig. 1g), or from a uniform distribution over all cells (Fig. 1h). This post-selection is possible due to our tracking system, and it is a differentiating aspect between active particle and electronic systems.

We study devices built out of $(L_x, L_y) = (12, 6)$ and $(L_x, L_y) = (13, 14)$ unit cells in the horizontal ($x$) and vertical ($y$) directions, with either a small or a large density of injected active particles (see "Methods"). In the main text, we focus on the larger device, with $(L_x, L_y) = (13, 14)$, see Fig. 1. All other devices and the corresponding results are shown in Supplementary Discussion 2.

Because of the irregular microchannel walls and the collisions between particles, we model the collective motion of active particles as a Brownian motion in a stochastic network with transition probabilities between the cells introduced in Fig. 1e. Cells are represented by nodes in our model, depicted in Fig. 2a, b. Neglecting correlations between particles due to collisions, that we refer to as *jamming*, the continuous-time Markov master equation governs the probability distribution of the particles in time as[47]

$$\tau \frac{\partial \mathbf{P}}{dt} = \hat{W} \cdot \mathbf{P}, \tag{1}$$

where $\mathbf{P} = P_{ij\sigma}$ is the probability to observe a particle on site $(\sigma, i, j)$, $\sigma \in \{A, B\}$ is the sub-cell index, and $(i, j)$ are the primitive-cell

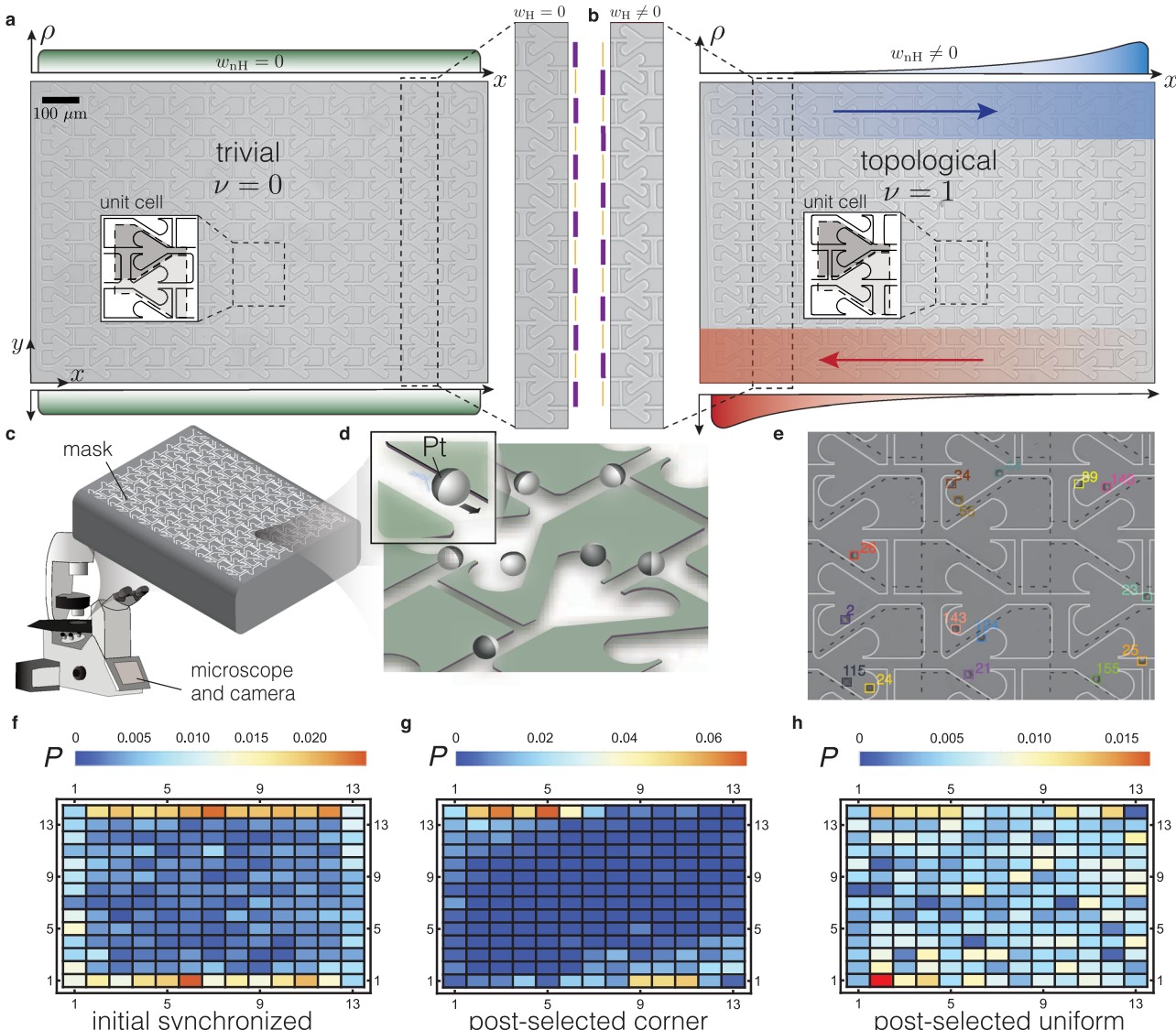

**Fig. 1 Experimental designs.** Panels (**a** and **b**) show the trivial and the topological devices, respectively, with $(L_x, L_y) = (13, 14)$ unit cells. The insets depict the unit cells which are the same for both devices except the vertical wide and narrow channels are reversed, as shown in the zoomed central inset and emphasized with orange and purple lines. The topological device displays chiral edge modes at the top and bottom, sketched in blue and red, respectively, and which are related to a non-vanishing Hermitian topological invariant, $w_H$. These topological edge modes have a non-vanishing non-Hermitian winding number, $w_{nH}$, responsible for the accumulation of active particles at the corners. This accumulation at the corners is expected to vanish for the trivial device, see the uniform density in green, because the topological invariant $\nu = w_H w_{nH}$ vanishes. **c**, **d** A schematic of the experimental setup, where Pt-coated $SiO_2$ Janus particles self-propel when hydrogen peroxide is added, following the topographic features of each design. **e** A portion of the device to illustrate the tracking of particles by using a neural network, each particle is uniquely identified within our algorithm. We locate particles within the cells outlined by the dashed lines. **f–h** Probability distribution data, **P**, of particles on the lattice when trajectories are tracked and synchronized to start at the same initial time (**f**), post-selected to start at opposing corners (**g**), or post-selected to have an initial uniform distribution (**h**). The particle distributions in (**g** and **h**) are shown 3 min after synchronization. These initial particle distributions are similar for both topological and trivial devices.

indices (see Fig. 2a, b). $\tau$ is the timescale for a particle to move between adjacent sites. $\hat{W}$ is the transition rate matrix (written explicitly in Supplementary Discussion 1). It depends on the four transitions probabilities: $t_\pm$ for the motion along or against the ratchet-like microchannels and, $t_1$ and $t_2$ for the motion along the wide and narrow microchannels, respectively. The timescale $\tau$ and the conditional probabilities are extracted from our experimental data in Supplementary Discussion 1. $\tau$ is roughly ten seconds and since it depends on various factors, from the concentration of chemicals to illumination, in practice we compute its probability distribution for each experiment. Also, we obtain $(t_1, t_2, t_+, t_-) \approx (0.20, 0.13, 0.52, 0.15)$. The large ratio $t_+/t_-$ shows the motion is

unidirectional on the horizontal axis and $t_1/t_2 \approx 1.5 \neq 1$ confirms that we can explore the difference between trivial and topological devices.

**Topological chiral edge motion.** We first use the post-selection technique to explore the edge dynamics of an ensemble of trajectories synchronized to start either at the top left or the bottom right corners (see Fig. 1g). We select 281 trajectories, of about 17 min each, for the trivial device, and 327 trajectories, of about 16 min each, for the topological case.

In Fig. 2c, d, we show the density of active particles as a function of time for trivial (c) and topological (d) devices.

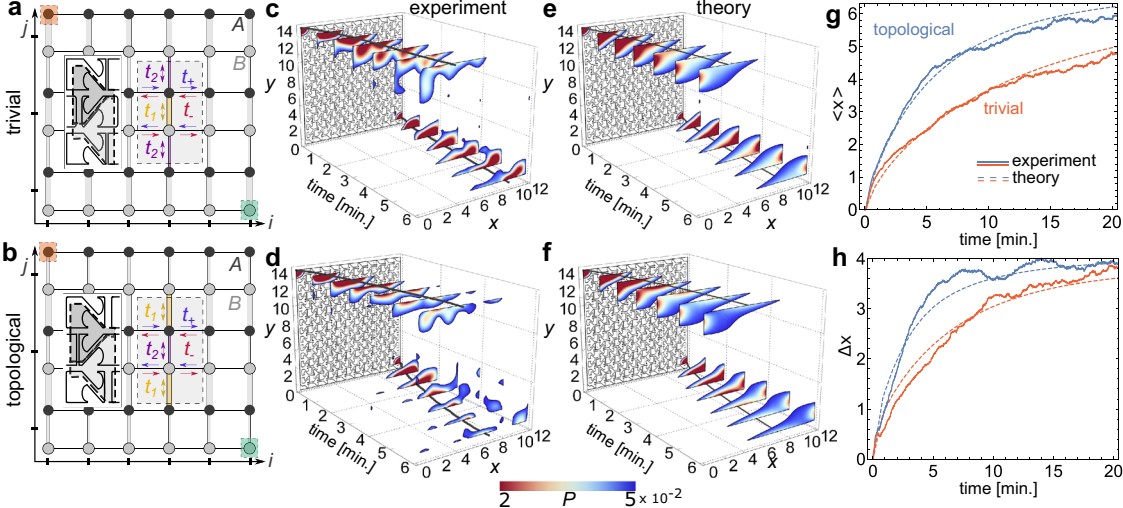

**Fig. 2 Chiral edge motion. a, b** Illustration of the stochastic network in Eq. (1) on which we superimpose the unit cell of the experimental device (see Fig. 1 a, b). Each dot corresponds to a cell of the device and each link represents one of the transition probabilities ($t_1, t_2, t_+, t_-$) to move in each cardinal direction (see main text). The unit cell of this lattice is the rectangle in gray, which contains black and gray nodes that are labeled by the sub-cell indexes $A$, $B$, respectively, introduced in Eq. (1). **c–f** Propagation of particles starting from the top left and bottom right corners (the orange and green areas in **a, b**) for different times, for the (**c, e**) trivial and (**d, f**) topological device. We show both the (**c, d**) experimental and (**e, f**) theoretical behavior for $(L_x, L_y) = (13, 14)$. The solid lines in (**c–f**) follow the average position of the particles starting from top left and bottom right corners. **g** Average position and (**h**) standard deviation in units of the cell index of a set of trajectories starting from either top left or bottom right corners of the device, which we compare with our model (see also Supplementary Discussion 1). The active particles of a topological device propagate faster than in a trivial device.

Qualitatively, the particle distribution propagates unidirectionally along the edge, faster in the topological device than in the trivial one. Quantitatively, the time-dependent average displacement $\langle x \rangle$ and spread $\Delta x \equiv \sqrt{\langle (x - \langle x \rangle)^2 \rangle}$ for particles starting from the top left corner, confirms this behavior, see Fig. 2g, h. We observe that active particles in the topological device are ahead of the trivial device, by about one unit cell after 3 min.

The motion of active particles is understood decomposing Eq. (1) into the normal modes, $\mathbf{P_k}$, defined by

$$\hat{W} \cdot \mathbf{P_k} = \lambda_\mathbf{k} \mathbf{P_k}, \qquad (2)$$

where $\lambda_\mathbf{k}$ is a complex scalar which depends on wavevector $\mathbf{k} = (k_x, k_y)$ for periodic lattices. The real part of $\lambda_\mathbf{k}$ sets the lifetime of the normal mode, and its curvature $\partial_k^2 \mathrm{Re}(\lambda_\mathbf{k})$ at $\mathbf{k} = 0$ sets its diffusion coefficient. The slope of the imaginary part $\partial_k \mathrm{Im}(\lambda_\mathbf{k})$ at $\mathbf{k} = 0$ sets the velocity of the normal mode (see Supplementary Discussion 1). For open boundaries along $y$ and periodic along $x$, the spectrum $\lambda_\mathbf{k}$ can be represented as a function of $k_x$. The boundary condition is such that the probability distribution vanishes outside the lattice. The resulting spectrum is shown in Fig. 3a, b for the topological device and c, d for the trivial one. The spectrum is colored according to the localization of the normal modes, where red and blue colors denote states at the top and bottom edges, respectively. The normal modes localized at the edge govern the propagation of a particle distribution localized at an edge. Some edge modes have a chiral group velocity, shown by the slope of the imaginary part of the spectrum at $k_x = 0$ in Fig. 3b, and are absent for the trivial device (see Fig. 3c, d). If in addition to enforcing a vanishing probability distribution outside the lattice we impose that detailed balance is preserved at the boundary, an additional edge potential partially hybridizes the edge modes with bulk modes, but does not remove them (see Supplementary Discussion 1).

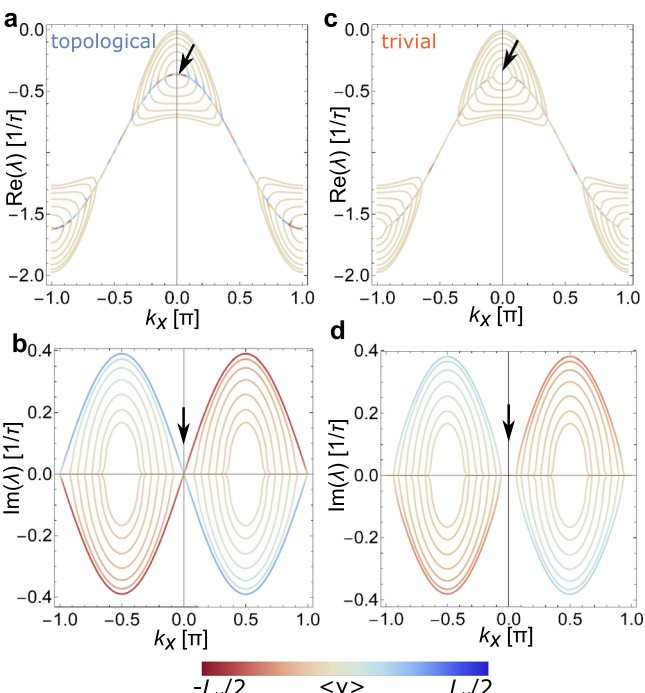

**Fig. 3 Stochastic chiral edge states.** Real (**a, c**) and (**b, d**) imaginary parts of the normal modes of the rate matrix $\hat{W}$ for open-boundary conditions in the $y$-direction, for the (**a, b**) topological and (**c, d**) trivial devices. The color denotes the average $\langle y \rangle$ position of a normal mode. Modes in green are delocalized and correspond to bulk states. Modes in blue and red correspond are strongly localized on the top or bottom edges, respectively. The arrows in (**c, d**) highlight the presence or absence of topological edge modes close to $\mathbf{k} = 0$, where their lifetime $\mathrm{Re}(\lambda)$ is largest.

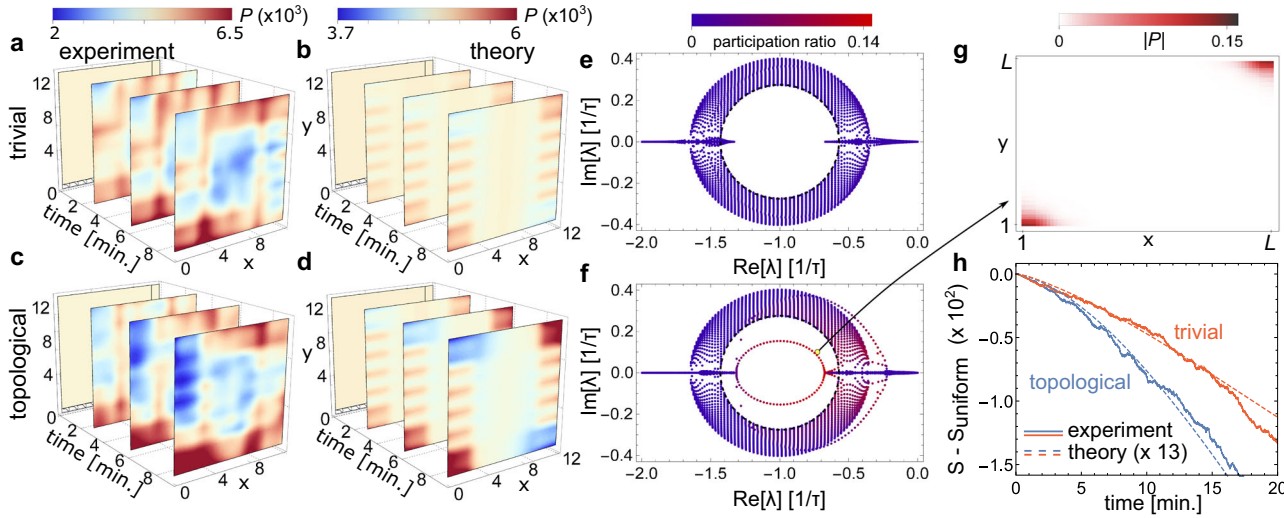

**Fig. 4 Second-order non-Hermitian skin effect of the active particles. a–d** We compare the distribution of particles in the trivial (**a**, **c**) and topological (**b**, **d**) devices with $(L_x, L_y) = (13, 14)$, observed experimentally (**a**, **b**) and predicted theoretically (**c**, **d**). We observe that more particles locate on the top right and bottom left corners. **e**, **f** Parametric representation of the real and imaginary parts of the spectrum of normal modes for the model of the trivial (**e**) and topological (**f**) devices with $(L_x, L_y) = (L, L) = (70, 70)$, colored with the participation ratio of each normal mode $\sum_{\sigma ij} |P_{\sigma ij}|^4$, which is small for a delocalized mode. The gap in the periodic band structure is within the dashed circle (see Supplementary Discussion 1). The number of localized modes within the point gap is proportional to the size of the device, $L$, and are localized at the corners as shown in (**g**). **h** Shannon entropy of the particle distribution over time. The smaller entropy in the topological device is associated with an accumulation of particles at the corners, signaling the second-order non-Hermitian skin effect. The theoretical figures match the experimental curves when multiplied by a factor ×13, suggesting that particle jamming, that occurs frequently for higher densities of active particles, contributes to enhance the accumulation.

The existence of edge modes is guaranteed by a bulk Hermitian topological invariant. In the vertical direction the couplings, $t_1$ and $t_2$, alternate between weak and strong (see Fig. 1b). This is the stochastic equivalent of the Su–Schrieffer–Heeger model of the polyacetylene chain[53], which is characterized by a winding number $w_H$ in the $y$-direction (see Supplementary Discussion 1). $w_H = 1$ and $w_H = 0$ for the topological and trivial device models, respectively. This implies that they, respectively, have, or not, topological edge modes for a boundary in the $y$-direction, indicated by black arrows in Fig. 3b, d.

The topological edge modes have a noticeable effect on the edge particle dynamics of the active particles. We probe them by initializing Eq. (1) with a probability distribution localized at the top left corner. We compare the experimental results with theoretical predictions in Fig. 2e–h, using the parameters $t_1, t_2, t_+, t_-$, and $\tau$ set by our statistical analysis of the experimental data. The theoretical curves qualitatively reproduce the experimental trends without any fitting parameter, for both topological and trivial devices. This demonstrates that the displacement in the topological device is larger as a consequence of the topological edge modes.

The above observations are reproduced for smaller devices and larger densities of active particles, see Supplementary Discussion 2. We find that for larger densities, the particles are slower than theory predicts, an effect we attribute to particle jamming, neglected in our model. A lower density of particles prevents jamming, in which case the motion compares better with our model.

**Corner accumulation from second-order non-Hermitian skin effect.** The detailed balance of the unit cell implies that $\hat{W}$ has no strong topological invariant[54] (see Supplementary Discussion 1). Moreover, $\hat{W}$ has inversion symmetry, implying that the first-order skin effect vanishes.

To derive the second-order skin effect of $\hat{W}$ we consider the topological edge modes, $\mathbf{P}_{s,\chi}$, at the top ($\chi = +$) and bottom

($\chi = -$) described by a 1D equation $\tau \partial_t \mathbf{P}_{s,\pm} = H_{s,\pm} \mathbf{P}_{s,\pm}$ (see Supplementary Discussion 1). These modes locate on either $A$ (for $\chi = +$) or $B$ (for $\chi = -$) sub-lattice, have a ballistic propagation $\langle x \rangle_\pm = \pm (t_+ - t_-) t/\tau$, and diffuse by an amount $\Delta x = \sqrt{(t_+ + t_-) t/\tau}$ after a time $t$ (see Supplementary Discussion 1). The real part of $H_{s,\pm}$ is finite and indicates that the contribution of the topological edge modes to the total probability decays over a timescale $\tau_d = \tau/(t_1 + t_2)$. $\tau_d$ sets how far the particles propagate due to the topological edge modes.

The edge modes $H_{s,\pm}$ have a finite 1D winding number $w_{nH} = \pm 1$ that implies a 1D non-Hermitian skin effect[21–23]. Since the edge modes are spatially separated, active particles can accumulate at the top and bottom corners. We detect the accumulation of active particles experimentally by post-selecting trajectories that start from a uniform configuration (see Fig. 1h and "Methods"). We observe an accumulation of active particles at the corners which is larger in the topological device (Fig. 4a, c) and that qualitatively compares with our model (Fig. 4b, d).

This observation can be made quantitative using the Shannon entropy of the particle distribution $S = -\sum_{ij\sigma} P_{ij\sigma} \ln(P_{ij\sigma})$. The entropy is maximal for a uniform distribution of particles, $S < S_{\text{uniform}} = \ln(L_x L_y)$, and it decreases if particles localize. We average out other sources of particle localization unrelated to the non-Hermitian skin effect by averaging the probability distribution over neighboring cells (see Supplementary Discussion 2). The experimental and theoretical entropies are depicted in Fig. 4h. Both figures show a smaller entropy in the topological device than in the trivial one. They depart from each other at the same rate, yet the absolute values of the experimental entropies are a factor 13 smaller than theory. Our model thus captures the difference between trivial and topological devices, but underestimates the accumulation that occurs in the experiment. A potential reason is that, in the experiment, a local increase in the number of active particles leads to particle jamming, neglected in our model. When we decrease the density of particles to reduce

jamming, the entropy still drops but is similar for topological and trivial devices. This suggests that there is a critical density of particles to observe corner accumulation.

The corner accumulation we observe is a consequence of the chiral motion of topological edge modes, that accumulate at the corners for long times. In the topological device, the chiral edge motion occurs because $w_H \neq 0$, and the corner accumulation occurs because $w_{nH} \neq 0$ for $H_{s,\pm}$. The two devices thus have a different topological invariant

$$\nu = w_H w_{nH}, \tag{3}$$

which equals one or zero for the topological and trivial devices, respectively. The topological invariant (3) highlights the coupled-wire nature of our system, and is reminiscent of weak topological phases. This observation motivates different viewpoints on the higher-order topological behavior we observe, as we discuss in the Supplementary Discussion 1.G. In Supplementary Discussion 1.F, we show that this invariant is equivalent to that in ref. [33], and that it signals a second-order skin effect as follows. First, $\hat{W}$ has inversion symmetry, and a point gap spectrum in which corner modes appear only for open-boundary conditions in both directions (Fig. 4f, g). Second, the number of corner modes scales as the edge length $L$, rather than the system size $L^2$, a defining characteristic of the second-order the skin effect[34].

Our work establishes a strategy to design circuits that spontaneously break detailed balance to guide and accumulate active matter, enforced by robust out-of-equilibrium topological phenomena. This design strategy and the 3D printing capabilities, permit to envision 3D extensions where the accumulation of particles is guaranteed by non-planar surfaces (e.g., using ramps, or different levels)[55] or by implementing dynamic topographical pathways topology[56].

## Methods

**Particle preparation**. Crystal slides of $25 \times 25$ mm are sequentially cleaned with acetone and isopropanol in a sonication bath for 2 min. The glass slides are then dried with compressed air and treated with oxygen plasma for 10 min. Commercial silica microparticles (Sigma-Aldrich) of 5 μm diameter size are deposited on the glass slides by drop-casting and left to dry at room temperature. We then sputter a 10-nm layer of Pt (Leica EM ACE600) to integrate the catalytic layer on the silica microparticles. The samples are kept in a dry and closed environment. The Janus particles are released for each experiment after being briefly exposed to an Ar-plasma in order to increase their mobility. For each experiment, we dilute particles from a third of a glass slide in 1 mL of water via sonication for a few seconds, after what they are ready to be used.

**Microfabricated model substrate for topological guidance: design and fabrication**. The microscale features that define the masks are designed and microfabricated on a silicon wafer, and later transferred by replication to a thin structure of polydimethylsiloxane (PDMS). The microchannel design is created with a computer-aided design (CAD) software (AutoCAD, Autodesk) and is made of ratchet-like structures that favor directional trajectories of the active particles[46]. Direct writing laser lithography (DWL 66FS, Heidelberg Instruments) is used to produce an AZ® resist master (AZ 1512HS, Microchemicals GmbH) of 1.5-μm thickness on the silicon wafer. A PDMS replica from the rigid mold/master is produced to obtain the open channel microfluidic device with the desired features. For reusability purposes, the master is silanized with trichloro (1H,1H,2H,2H-perfluorooctyl)sylane (Sigma-Aldrich) by vapor phase for 1 h at room temperature, this operation reduces the adhesion of PDMS to the substrate. To obtain a PDMS thin layer with the inverse pattern from the rigid mold (so-called PDMS replica), the PDMS (Sylgard 184, Dow Corning) monomer and cross-linker are mixed at a ratio of 10:1 and degassed for 1 h. Afterward, the solution is spin-coated at 1000 rpm for 10 s on the top of the master and cured for 4 h at 65 °C. The replica is carefully released, obtaining the desired open microchannel designs with the sub-micrometer step-like topographic features that allow the topological guidance of active particles.

**Setup fabrication**. A circular well of 2-mm height and 1.5-cm diameter is designed with a CAD software and post-treated with Slic3r and Repetier softwares in order to generate the G-code required for 3D printing it with a Cellink's Inkredible+ 3D printer. Polydimethylsiloxane (PDMS) SE1700 cross-linker and monomer are mixed at a ratio of 1:20, and the viscous solution is added to the cartridge used in

the 3D printer. The pneumatic extrusion of the 3D printer allows to print the PDMS-based structures on clean crystal slides ($60 \times 24$ mm) at a pressure of 200 kPa. Once the well structure is printed, the slides are left overnight at 65 °C for curing, to obtain a stable structure. To ensure good sealing of the 3D-printed well, a mixture of 1:10 PDMS Sylgard 184 is added around the outer side of the well, followed by a curing period of 4 h at 65 °C.

The glass slides with the 3D-printed wells are sequentially cleaned with acetone and isopropanol, and dried with compressed air. The glass substrate is activated by exposure to oxygen plasma for 30 s. The microratchet-like patterns in the section "Microfabricated model substrate for topological guidance: design and fabrication" are then immediately transferred to the wells to ensure good attachment of the patterned PDMS to the glass. The whole setup is ready and it is kept in a dry and closed environment until use. Before use, the setup is cleaned with a sequential wash of acetone and isopropanol, and exposed to oxygen plasma for 30 s. The setup design is such that it minimizes the accumulation of gas bubbles at the center of the sample which permits long-time experiments, from 30 min to 1 h.

**Active particles concentration**. We perform our experiment for two different concentrations of active particles. This allows to compare how the density of active particles qualitatively affects our experiment. Indeed, a larger density of particles enhances the Brownian motion, due to the many collisions, but also decreases the velocity of particles and leads to more clusters. The two densities are obtained by injecting particles in either one of the two solutions:

- *High density of active particles*. We dilute 17.5 μL of the solution of Janus particles prepared as detailed in the "Methods" describing particle preparation with 35 μL of $H_2O_2$ at 2% per volume and 47.5 μL of water.
- *Low density of active particles*. We first dilute 300 μL of the solution of Janus particles prepared as detailed in the "Methods" describing particle preparation with 700 μL of water. Then, we dilute 17.5 μL of this solution with 35 μL of $H_2O_2$ at 2% per volume and 47.5 μL of water.

This way we expect a ratio of concentrations of a third to a half between high and low densities of active particles.

**Tracking**. In order to track a large amount of particles, we have developed a Python script based on a neuronal network detection algorithm[52]. This neuronal network is taught to recognize a particle from a video frame and to discard anything else. Once trained, we use this neuronal network to detect particles on every frame of the video. The experiments are recorded using a contrast microscope (Leica) with a ×10 objective at 2 frames per second. A particle fits in a box of 8–9 px$^2$ on the resulting video. Our software relates the detected particles between frames to find the trajectory of a given particle. The following paragraphs detail the procedure to train the neuronal network and to construct trajectories from the particle we detect on each frame.

To train the neuronal network, we construct a dataset of images that show one or many particles. We perform this step manually by cropping a square image from a video frame. This cropped image is centered close to a particle's center, keeping the selected particle within, and is three times the size of a particle. We also draw a square region of the size of the particle on this cropped image, like the squared, colored, boxes in Fig. 1e, that indicate the location of the particle on the cropped image. We construct our dataset by applying this procedure on many particles. We extend this dataset by also adding cropped images that do not contain any particle. This way, our dataset is composed of 8700 images with particles and 8700 images that do not show any particle. We input this dataset to a neuronal network based on tiny YOLOv3[52]. We simplify the network by transforming the images to 8-bit grayscale and considering there are only two sizes for the boxes surrounding the particles. We train the network for a day and obtain a mAP-50 larger than 95%, which is sufficient for our purpose. This generates a configuration file that we use to detect particles of an image. We illustrate the output of this detection algorithm in Fig. 1e, where an identifier is assigned to each detected particle.

To build the trajectory of a particle located at position $\mathbf{r}_t$ at time $t$, we estimate that its location $\mathbf{r}_{t+1}$ at a time $t + 1$ is

$$\mathbf{r}_{t+1} = \mathbf{r}_t + \frac{\mathbf{r}_t - \mathbf{r}_{t-1}}{2}, \tag{4}$$

where the second term is an estimate of the displacement of the particle, which is zero for the first frame since we miss the position $\mathbf{r}_{-1}$. We use this position estimate to crop a square image of the frame $t + 1$ centered on $\mathbf{r}_{t+1}$ and twice the size of a particle. We apply our neuronal network detection algorithm on this cropped image to detect all particles it may contain. We calculate the distance between each detected particle and each of the particles we are tracking at time $t$, keeping always the lowest distance calculated and the identification number (ID) of the associated particle. We create a list with these IDs and distances. First, we filter by selecting only those distances that have the same ID as the particle we want to detect. If any, we will chose as $\mathbf{r}_{t+1}$ the closer object detected. If we cannot find the same ID, we select $r_{t+1}$ as the closest object detected to this cropped image. If finally, we have not detected any particle, we set $\mathbf{r}_{t+1} = \mathbf{r}_t$. We apply this algorithm for a set of particles we manually select at the initial frame, as well as particles that may enter into the frame while recording. This tracking procedure allows to track particles even when they collide; only clogs of 5 particles or more may result in particle loss.

Particles may leave the sample; if this happens their position is fixed to that frame and is removed after a period of inactivity. Also particles can adhere electrostatically to the PDMS, or to each other, or be inactive. Therefore we end the trajectory if the particle is inactive for a long period of time. We use GPU acceleration for the neuronal network detection, based on the latest NVIDIAĎ¢™ drivers.

The filtered trajectories are given in $x,y$ pixel coordinates, which we map to the cell coordinate of our model. This transformation is obtained after aligning the axes of the recorded videos and decomposing the image over the lattice structure in Fig. 1e, shown with dashed lines. We then assign each of the now aligned $(x,y)$ coordinate to a cell index, using the Python library *shapely*.

**Post-selection of trajectories**. We organize the recorded trajectories in a database for each experimental configuration. We only keep track of the position in the unit-cell index to eliminate any intrinsic motion of a particle within a cell and thus to reduce noise. In the manuscript, we consider two post-selected configurations: (1) where particles start at the top left and bottom right corners (see Fig. 1g) and (2) where particles start from a uniform distribution (see Fig. 1h) over the lattice. The two post-selection procedures are as follows:

- *Post-selected corner*. We select particles starting from the corners in two steps. We first select trajectories that go through one of the two corners. Then, among the selected trajectories, we remove all frames before the one where the particles first enters one of the two corners. This ensemble of trajectories is what we use for the analysis in Fig. 2e–h.
- *Post-selected uniform*. We select particles uniformly scattered over the sample based on simulated annealing. We focus on the probability distribution $P_{\sigma ij}(t=0)$ of all trajectories on their first frame. We scan trajectories from the longest to the shortest and remove the first frames of a trajectory until the newly generated distribution of particles $P'_{\sigma ij}(t=0)$ is more uniform than the one, $P_{\sigma ij}(t=0)$, at the previous step. We consider the distribution is more uniform when

$$C(P') = \sqrt{\sum_{\sigma ij}\left(P'_{\sigma ij}(t=0) - \frac{1}{L_x L_y}\right)^2} < C(P). \qquad (5)$$

During this procedure, we remove trajectories that have <6 min of video to remove noise in the first frames. This ensemble of trajectories is what we use for the analysis in Fig. 4.

## Data availability
Supplementary Information is available for this paper. The data that supports our finding is available on Zenodo[57]. Additional data are available upon request.

## Code availability
The code used to generate the figures is available upon request.

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

## Acknowledgements

A.G.G. and S.T. are grateful to M. Brzezinska, M. Denner, T. Neupert, Q. Marsal, S. Sayyad, and T. Sépulcre for discussions. A.G.G. and S.T. acknowledge financial support from the European Union Horizon 2020 research and innovation program under grant agreement No. 829044 (SCHINES). A.G.G. is also supported by the ANR under the grant ANR-18-CE30-0001-01 (TOPODRIVE). L.P. is grateful to J. Katuri for discussions about ratchet design and to J. Fuentes for the PDMS wells fabrication. L.P. acknowledges financial support from MINECO for the FPI BES-2016-077705 fellowship. M.G. thanks MINECO for the Juan de la Cierva fellowship (IJCI2016-30451), the Beatriu de Pinós Programme (2018-BP-00305), and the Ministry of Business and Knowledge of the Government of Catalonia. I.P. acknowledges support from Ministerio de Ciencia, Innovación y Universidades (Grant No. PGC2018-098373-B-100), DURSI (Grant No. 2017 SGR 884), and SNF (Project No. 200021-175719). S.S. acknowledges the CERCA program by the Generalitat de Catalunya, the Secretaria d'Universitats i Recerca del Departament d'Empresa i Coneixement de la Generalitat de Catalunya through the project 2017 SGR 1148 and Ministerio de Ciencia, Innovación y Universidades (MCIU)/ Agencia Estatal de Investigación (AEI)/Fondo Europeo de Desarrollo Regional (FEDER, UE) through the project RTI2018-098164-B-I00. S.S. acknowledges financial support from the European Research Council (ERC) under the European Union's Horizon 2020 research and innovation program (grant agreement No. 866348). All the authors acknowledge MicroFabSpace and Microscopy Characterization Facility, Unit 7 of ICTS "NANBIOSIS" from CIBER-BBN at IBEC for their support in the masks design and fabrication.

## Author contributions

M.G. designed the microchannel designs and PDMS wells. L.P. fabricated the micro-channel devices, performed the experimental work, and extracted the corresponding data, developing also a neural network-based tracking system to evaluate the trajectories of the active Janus particles. L.P., S.T., and A.G.G. analyzed the experimental data, with input from I.P. and S.S.S.T. derived the theoretical model and computed its observables. S.T. and A.G.G. wrote the manuscript, with input from all authors. A.G.G. devised the initial concepts for the experimental setup and for the theoretical modeling, and supervised the project.

## Competing interests

The authors declare no competing interests.
