## [Peer Review File · Nature Communications]

Editorial Note: This manuscript has been previously reviewed at another journal that is not operating a transparent peer review scheme. This document only contains reviewer comments and rebuttal letters for versions considered at Nature Communications .

REVIEWERS' COMMENTS

Reviewer #1 (Remarks to the Author):

I greatly appreciate the authors' reply and revisions. Given that, my overall evaluation still remains the same: this work observed the effect that can be considered as a simplistic union of non-hermitian and higher-order topology rather than a fundamentally new phenomenon and, although it may show some promise for technological advance, this point is not well-justified in the manuscript due to the lack of concrete demonstration.

In particular, the authors basically agree with my comment 1 and, although they emphasize the unbroken inversion symmetry, (in my opinion) this point is rather obvious by construction because the directional 1D chains are arrayed in a staggered manner with opposite directions in the present setup. This is also what has been pointed out in my comment "in the limit $t_1 \rightarrow 0$, ... Because the top and the bottom 1D edges are completely decoupled from the bulk, they simply form two 1D chains associated with the asymmetric hoppings (with different chirality). Because of this asymmetry, modes will localize at one side of 1D chains, which of course correspond to the "corners" of the original 2D system in an obvious way. This is nothing but the ordinary non-Hermitian skin effect".

In this sense, it also seems to me that the authors' reply to my comment 2 is not very satisfactory. The authors also emphasize a potential promise towards extending the experimental platform to the 3D cases. I agree that such an extension would be very interesting if possible, however, this is not what has been achieved in the experimental setting developed here.

In view of all these points and the revisions made, I feel that the observed effect is a phenomenon that can be interpreted as a rather straightforward extension of the non-Hermitian skin effect. The observations of the skin effect are already reported in several papers like Nat. Commun. 10, 4608 (2019); Phys. Rev. Research 2, 023265 (2020); PNAS 117, 29561 (2020). Given that, I have to say that I am not entirely sure whether or not progress made by the current manuscript is significant enough to meet the high bar set by Nat. Commun.

Reviewer #2 (Remarks to the Author):

I reviewed this manuscript for Nature before, and I recommended it for publication but for a bit of a lower tier journal. I believe that Nature Communications is an appropriate venue. The changes made by the authors are acceptable in my opinion.

Overall I would like to congratulate the authors for a very nice study

Reviewer #3 (Remarks to the Author):

All my comments have been addressed well.
In my opinion, the paper is ready for publication.

I. REPLY TO REFEREE #1

We thank the referee for their appreciation of the previous update of our manuscript. We indeed agree that our platform gives an intuitive understanding of the second-order non-Hermitian skin-effect but we disagree with the assesement that it narrows it down to a special case of the first-order non-Hermitian skin-effect. In particular, the Referee writes

In particular, the authors basically agree with my comment 1 and, although they emphasize the unbroken inversion symmetry, (in my opinion) this point is rather obvious by construction because the directional 1D chains are arrayed in a staggered manner with opposite directions in the present setup. This is also what has been pointed out in my comment "in the limit $t_1 \rightarrow 0$, ... Because the top and the bottom 1D edges are completely decoupled from the bulk, they simply form two 1D chains associated with the asymmetric hoppings (with different chirality). Because of this asymmetry, modes will localize at one side of 1D chains, which of course correspond to the "corners" of the original 2D system in an obvious way. This is nothing but the ordinary non-Hermitian skin effect".

This argument is strictly valid for $t_1 = 0$, where the device is a collection of 1D wires without inversion symmetry. In our experimental setup, this idealistic situation does not occur and the device is a 2D system with inversion symmetry, a situation that prevents the first-order non-Hermitian skin-effect. On more general grounds the argument of the referee, applied to ordinary topological phases, would minimize or trivialize the importance of crystalline topological phases, such as weak topological insulators, as they can also be understood in the coupled-wire construction. We believe it is hard to argue against how useful this construction is, if only judging by the volume of research in crystalline topological phases, precisely because understanding topological phases becomes simple in certain limits of the parameters, as the one mentioned by the Referee.

Specifically, it is known that higher-order topological insulators with topologically protected corner states can be understood as a superposition of lower dimensional strong topological phases. We consider that the argument of the referee trivializes the interpretation of several experiments in higher-order topological phases, including those in bismuth, recently interpreted as a higher order topological insulator [see DOI:10.1038/s41567-018-0224-7]. The situation we report proceeds very similarly to a higher-order topological phase, as can be seen from our topological invariant ν in Eq. (3) which is a product of two 1D invariants.

Lastly, we believe the Referees' argument is unfair to our experimental observation of corner mode accumulation. The fact that there is a theoretically simple limit does not guarantee that the experiment will actually succeed in localizing particles and that said localization is detectable.

Given the above, we understand the intuitive picture provided by the referee but we also consider it to be an unfair oversimplification of higher order topological phases. In order to contextualize our findings we have added the section Supplementary Information 1.G where we put into perspective the second-order non-Hermitian skin-effect with both higher-order topological phases and the first-order non-Hermitian skin effect.

Nonetheless we would like to thank the referee for their constructive and detailed criticism which has made our work better. We hope that with our clarifications and changes, our work is now suitable for publication in Nature Communications.

Changes: We refer to the newly added Supplementary Information 1.G at lines 323-326 of the main text. Supplementary Information 1.G is at page 10 of the Supplementary Material. We have added a reference to F. Schindler et al. "Higher-order topology in bismuth" doi:10.1038/s41567-018-0224-7 (2018).

II. REPLY TO REFEREES #2 AND #3

We thank the referees for their positive replies on our manuscript. Their comments have been helpful in improving the overall quality of our manuscript.